# Implicit Negativity Bias Leads to Greater Loss Aversion and Learning during Decision-Making

**DOI:** 10.3390/ijerph192417037

**Published:** 2022-12-19

**Authors:** Francisco Molins, Celia Martínez-Tomás, Miguel Ángel Serrano

**Affiliations:** Department of Psychobiology, Universitat de València, Av. Blasco Ibáñez, 13, 46010 Valencia, Spain

**Keywords:** negativity bias, loss aversion, reinforcement-learning, decision-making, Iowa Gambling Task

## Abstract

It is widely accepted there is the existence of negativity bias, a greater sensitivity to negative emotional stimuli compared with positive ones, but its effect on decision-making would depend on the context. In risky decisions, negativity bias could lead to non-rational choices by increasing loss aversion; yet in ambiguous decisions, it could favor reinforcement-learning and better decisions by increasing sensitivity to punishments. Nevertheless, these hypotheses have not been tested to date. Our aim was to fill this gap. Sixty-nine participants rated ambiguous emotional faces (from the NimStim set) as positive or negative to assess negativity bias. The implicit level of the bias was also obtained by tracking the mouse’s trajectories when rating faces. Then, they performed both a risky and an ambiguous decision-making task. Participants displayed negativity bias, but only at the implicit level. In addition, this bias was associated with loss aversion in risky decisions, and with greater performance through the ambiguous decisional task. These results highlight the need to contextualize biases, rather than draw general conclusions about whether they are inherently good or bad.

## 1. Introduction

Most events and experiences in our daily living can be classified along a hedonic dimension, according to the positive or negative emotions they produce [1]. The emotional significance of a stimulus enhances its processing [2,3]. Therefore, this stimulus would have a greater influence on our perception, judgement, and decision-making. In addition, its valence (positive or negative) could provide an extra boost in that processing [1].

It is widely accepted there is the existence of negativity bias in human beings [4,5,6,7], referring to the greater sensitivity to negative stimuli compared with positive stimuli [4,8], and to the higher predisposition to consider ambiguous emotional stimuli as negative than positive [9]. The origin of this bias has been addressed from multiple perspectives (see Kanouse [10] for a review) and it is still an open question, however, its existence has been evidenced in both verbal and non-verbal stimuli [1,11]; in affective judgments [9], social information processing [12], during the child development [6], and on consumer behavior [13]; also when using event-related potentials [14], and peripherical physiological measures [15,16]. However, this generalizability was recently challenged. In their review, Kauschke et al. [1] concluded that this bias does not always arise and that, in fact, a positivity bias sometimes occurs, especially during childhood. Nevertheless, the meta-analysis of Joseph et al. [8], conducted in 874 samples and 53,509 participants, consistently revealed the presence of negativity bias. Therefore, current research points to negativity as the most widespread phenomenon, although it may be subject to variability depending on individual and contextual factors, such as age, stimulus modality, or task [1,9]. In addition, since some tasks only measure explicit or conscious responses, they could not be sensitive enough to capture the emotional bias; it is also important to address implicit automatic responses [9].

Following an ecological-rationality approach [17,18], classifying the negativity bias as advantageous or disadvantageous depends on the context. Focusing on decision-making and according to classical-rationality models, in risky contexts (where decision-rules are explicit and outcomes probabilities are known), individuals should take decisions strategically, following the rules of probability, logic, and maximizing the utility [19,20,21]. However, emotional biases can produce a jumping-to-conclusion effect that impairs this mathematical calculation and could lead to non-rational choices [20,22]. This is the case of the prominent loss aversion bias, whereby losses loom larger than gains [21,23]. So, losses have a greater psychological impact [24] and could produce ‘anomalies’, such as the framing or the endowment effect [23], that violate classical-rationality axioms just to avoid losses at any price. Recently, it has been proposed that loss aversion could be decomposed into the response bias and the valuation (or negativity) bias [25]. Consequently, those with a greater negativity bias should also express greater loss aversion and therefore make more biased decisions in risky decision-making contexts. This relationship between loss aversion and negativity bias was also theoretically stated by Kahneman [26], from the field of risky decision-making, and Kanouse [10], from the impression formation literature but, to our knowledge, it has not been empirically tested to date.

On the other side, under ambiguous decisions (compared to risky decision-making), i.e., when uncertainty is high and there exist several outcomes with unknown probabilities [20,27,28], people would not be able to follow strategies such as utility maximization [19] and would rely on the reward or punishment experiences after each decision. These experiences produce emotions that are linked to the different decision alternatives and act as somatic markers that guide following decisions [27,29]. Sensitivity to rewards and punishments plays a key role in this reinforcement-learning process [20]. In this case, having a greater negativity bias could enhance the effect of the punishments and would help to avoid those stimuli that produce them [3,30]. As learning research evidenced, this negative reinforcement would lead to faster learning [6,31]. Thus, it would be expected that having a higher negativity bias would be conducive to better decision-making under ambiguity since this bias could improve the reinforcement-learning. However, this hypothesis has never been tested to date.

Based on the above, the aim of our study is to provide new evidence of the existence of negativity bias, as well as to explore its role when risky and ambiguous decisions are made. This would shed light on the generalizability of the negativity bias and, on the other hand, help to better contextualize the adaptive/disadaptive role of this bias depending on the decisional environment. We hypothesize that, in a classification task of emotional faces [9], ambiguous faces will be more often classified as negative, showing the presence of the negativity bias. In addition, the higher level of negativity bias will predict the higher level of loss aversion when taking risky decisions, and a better performance on an ambiguous decisional task.

## 2. Materials and Methods

### 2.1. Participants

Based on the effect size found in a previous work [9], an a priori power analysis using G*Power indicated a requisite of 34 participants (η2p = 0.2, power = 80%, α = 0.05) to detect whether participants display negativity bias, both at the explicit and at the implicit level. Seventy students were recruited by the means of a non-probabilistic sampling method, by asking them during their classes in the University if they wish to participate in a study in exchange for academic credits. Those interested filled out a self-administered questionnaire to ensure that they met the following inclusion criteria when first contacted: not having any neurological or psychiatric diseases; not consuming drugs habitually; and not having experienced a highly stressful event in the last month. In addition, participants were asked to not perform extenuating exercise or take drugs or alcohol in the last 24 h, and not smoke or take stimulant drinks in the 2 h before the experimental session. One participant was eliminated due to technical issues. A total of 69 participants (age: M = 22.33, SD = 2.29; women: N = 52, (75.3%)) were finally included in the study.

### 2.2. Procedure

Experimental sessions were carried out between 15:00 p.m. and 19:00 p.m. and lasted approximately an hour. Participants were collected in the University hall and accompanied to the laboratory. The general procedure was explained (see Figure 1), and informed consent was signed. Before starting the protocol, participants fulfilled a short, self-administered questionnaire to control the consumption of psychoactive substances and stimulants. Then, they performed the Face Rating Task [9] to measure their negativity bias level. Five minutes later the Lottery Choice Task [32] was employed to measure loss aversion in a risky decision-making context, and the Iowa Gambling Task [33] was used to assess decision-making under ambiguity. Both tasks were counterbalanced among participants. This study was approved by the Ethics Research Committee of the University of Valencia in accordance with the ethical standards of the 1969 Declaration of Helsinki.

### 2.3. Face Rating Task (FRT)

The FRT was utilized to measure the negativity bias. This task included 16 faces (8 surprised, 4 happy, and 4 angry), each presented four times in randomized order, for a total of 64 trials. Faces (8 male and 8 female) were extracted from the NimStim standardized facial expression stimulus set [34] with the consent of its developers. Each trial was composed of a black fixation cross which appeared in the center of a white background for 500 ms, and a 500 ms face presentation. After that, participants indicated whether they thought the expression was positive or negative by clicking a start button at the bottom of the display and clicking one of the two response option buttons (positive or negative) in the upper left- or upper right-hand corner of the display. 

It was checked whether the happy and angry faces were correctly classified as positive and negative, respectively. On the other hand, it was compared whether the rate of surprised (or ambiguous) faces classified as negative was higher than that of ambiguous faces classified as positive. This corresponds to the explicit measure of the negativity bias. In addition, MouseTracker 2.83 software [35] was used to obtain a more sensitive measure of negativity beyond explicit valence ratings. This software tracked the mouse’s trajectory as participants determined the valence of ambiguous facial expressions. During a trial, the trajectory can reflect either a straight line (when participant’s mouse moves directly from the start button to the response), or it can show a curvature (when it is pulled toward the opposite response during the decision process). This curvature reflects the implicit competition between positive and negative ratings. Thus, the maximum deviation (MD) was obtained for ambiguous faces classified as positive (positive-MD), and for those classified as negative (negative-MD). MD quantifies the attraction toward the unselected response by measuring the largest perpendicular deviation away from the most direct trajectory to the selected response [9]. The greater the MD, the greater the competition of the alternative response. A higher positive-MD indicates a greater implicit negativity bias since it reflects that, although the positive rating is finally chosen, the automatic response tends towards negative rating. Moreover, a lower negative-MD also indicates a greater implicit negativity bias since it reflects that negative ratings are more automatic.

In sum, three variables of the negativity bias were included in the study: (1) explicit negativity bias, (2) positive-MD, and (3) negative-MD. The last two refer to the implicit level.

### 2.4. Lottery Choice Task (LCT)

The LCT [32] was employed to measure loss aversion in a risk context. In this task, participants decide in six lotteries whether they accept or reject the bet. In each lottery the profit is fixed at 6€ and the loss varied through the bets (from 2 to 7€), yielding a successively decreasing expected value for each lottery. Following Gächter et al. [32], loss aversion was scored as the gain/loss ratio obtained from the highest bet accepted. This ratio shows how big the potential gain must be in relation to the potential loss for someone to accept the bets. Thus, the higher the ratio, the greater the loss aversion. Loss aversion values usually reported in the literature are 2–2.5 [36,37], which indicates that gains have to be at least twice as large as losses to accept a bet. As Rabin & Thaler [38] noted, loss aversion is not the same as risk aversion (tendency to avoid risky choices), therefore, it would be reasonable to ask whether this task really measures loss aversion and not risk aversion. However, as Gächter et al. [32] pointed out, based on the arguments of Rabin & Thaler [38], since this task offers small-stake gambles, behavior cannot be explained by risk aversion, otherwise, when someone had to deal with choices that involved large amounts at stake, “absurd degrees of risk aversion” [32] (p. 8) would be observed.

### 2.5. Iowa Gambling Task (IGT)

Decision-making under ambiguity was evaluated through the computerized version of the IGT [33,39]. Participants should get the maximum benefit possible from over 100 consecutive decisions where they can win and lose money. They can choose from four decks of cards: two disadvantageous (A and B) and two advantageous (C and D). A and B provide large immediate gains, but large losses in the long run. C and D provide lower short-term gains, but lower long-term losses, so their choice leads to higher profits. After each decision, the participant receives feedback that can be used to adjust future decisions. Performance was assessed by calculating the Iowa Gambling (IG) index: selections of C and D minus selections of A and B. The higher the IG, the higher the performance. This index was calculated for the entire task (IGTOTAL), and in blocks of 20 trials to study the learning curve.

### 2.6. Statistical Analyses

Outliers’ presence was checked with the 2.5 standard deviations method and the Kolmogorov-Smirnoff with Lilliefors correction was used to check normality. Analyses included repeated measures ANOVAs to examine both the differences between ambiguous faces classified as negative and positive (explicit negativity bias), and differences between negative and positive-MD (implicit level). General linear models were also performed to study associations between the negativity bias, and both loss aversion and the IGT performance. Finally, as a complementary analysis, the sample was divided into two groups (high and low negativity bias) taking the median as reference. Their loss aversion level and IGT performance were compared between them through ANOVAs. The α significance level was set at 0.05 and partial eta square (η2p) symbolizes the effect size. All analyses were performed with IBM SPSS Statistics 25.

## 3. Results

### 3.1. Negativity Bias

Firstly, it was checked whether the clearly positive and negative faces had been properly classified. The accuracy for both positive and negative faces was almost perfect, 99.81% (SD = 1.05). On the other hand, a repeated measures ANOVA was carried out to study whether there were differences between the ambiguous faces classified as positive and negative. At the explicit level, no differences were observed (F(1, 68) = 1.09, *p* = 0.30, and η2p = 0.02); finding that ambiguous faces were classified as positive (M = 17.09, SD = 8.37) and as negative (M = 14.9, SD = 8.37) with a similar proportion. However, differences were found at the implicit level (F(1, 68) = 9.85, *p* = 0.003, and η2p = 0.14), with a higher positive-MD (M = 0.31, SD = 0.31) than negative-MD (M = 0.15, SD = 0.25). That is, ambiguous faces showed a greater deviation of the mouse towards the opposite response (negative rating) when they were classified as positive. However, when ambiguous faces were classified as negative, the mouse’s trajectory reflected a straight response without attraction towards the positive ratings (see Figure 2). Finally, Pearson’s correlations revealed that positive-MD and negative-MD were not related with each other (r(69) = 0.053, *p* = 0.67); but both markers were related to the overall percentage of ambiguous faces classified as negative. Specifically, the higher the positive-MD, the higher the percentage of ambiguous faces classified as negative (r(69) = 0.366, *p* = 0.002), and the higher the negative-MD, the lower this percentage (r(69) = −0.320, *p* = 0.009).

### 3.2. Negativity Bias and Loss Aversion

First, it was necessary to identify whether the sample had loss aversion. The average value obtained in the lottery choice task was 2.63 (SD = 1.48), which is very close to that usually reported in the literature (2–2.5). In addition, it was studied whether negativity bias predicted loss aversion. Both the percentage of ambiguous faces classified as negative (explicit level), and negative-MD (implicit level) showed no associations with loss aversion (*p*’s > 0.05). However, positive-MD was significantly associated with the level of this bias (β = 1.91, SE = 0.54, t = 3.52, *p* = 0.001, and η2p = 0.17); i.e., the greater the attraction for the opposing option when ambiguous faces were classified as positive, the greater the loss aversion. In addition, dividing positive-MD by their median, it was compared whether there were differences in loss aversion between those who showed the greater and those who showed the lower positive-MD. When the ambiguous faces were classified as positive, the group that showed more deviation towards the opposing response (greater positive-MD), also had higher loss aversion (M = 2.99, SD = 1.74) than the group with a lower deviation (M = 2.24, SD = 1.03); F(1, 67) = 9.37, *p* = 0.035, and η2p = 0.07.

### 3.3. Negativity Bias and Iowa Gambling Task (IGT) Performance

It was studied whether negativity bias predicted performance on the IGT. Having a greater or lesser negativity bias, either at the explicit or implicit level, showed no associations with the IGTOTAL (*p*’s > 0.05). Similarly, no association was found between the explicit measure of negativity bias and performance in any of the 5 blocks of the IGT (*p*’s > 0.05). However, negative-MD was significantly associated with performance in the second block (β = −4.02, SE = 3.27, t = −1.23, *p* = 0.045, and η2p = 0.08) and the third block (β = −5.46, SE = 3.15, t = −1.72, *p* = 0.006, and η2p = 0.11). That is, the more automatic the negative rating for ambiguous faces, the higher the performance in those blocks. 

Again, by dividing negative-MD by their median, it was studied whether participants with greater or lesser negative-MD differed in performance on IGT. Repeated measures ANOVA for the 5 blocks of the IGT, including the groups formed by dividing negative-MD as the between-subject factor, was carried out. It was found to be a main effect for the moment factor (the 5 IGT blocks), F(4, 64) = 9.43, *p* < 0.001, and η2p = 0.30; this indicated that performance varied throughout the task, regardless of the group. In addition, a significant interaction moment*negative-MD groups (greater and lower negative-MD) was observed (F(4, 64) = 4.15, *p* = 0.005, and η2p = 0.21) which indicated that this evolution was different for each group. As can be seen in Figure 3 and Table 1, when contrasting the performance of both groups in each IGT block, the group that most automatically rated the ambiguous faces as negative was also the one that performed significantly better in blocks 2 and 3, as well as showing a significant trend towards better performance in block 4. No other negativity bias variable reported significant results in relation to IGT performance.

## 4. Discussion

The aim of this research was to provide new evidence of the existence of negativity bias when processing emotional stimuli and, furthermore, to study how this bias influences decision-making depending on the context. Results evidenced the presence of the bias, although only at an unconscious level. In addition, this bias was associated with both a more biased risky decision-making and a better decision-making under ambiguity. These results will be discussed in depth below.

Regarding our first hypothesis, results showed the presence of the negativity bias when classifying ambiguous emotional faces, however, this evidence occurred only at an implicit level; mouse’s trajectory when rating these faces as positive showed a significant deviation towards the opposing response, indicating that, although the trend was corrected at the explicit level, the initial impulse was to classify faces as negative. Yet, faces rated as negative showed a straight trajectory that reflects the absence of opposition. On the one hand, as in Brown et al. [9], this indicates that negativity bias could remain hidden if a methodology that explores beyond the conscious response is not used. Therefore, many studies that argued the absence of this bias (for a review see 1), should be revisited using new methods that replicate or modify results obtained. On the other hand, it seems that although there was negativity bias, it was not strong enough to affect the conscious response when classifying faces. According to dual-processing approaches [22,40], this emotional bias may be corrected by top-down mechanisms managed by the neocortex. Therefore, even when the initial impulse was to classify ambiguous faces as negative, a balanced rating between positive and negative valences was finally made. Nevertheless, individual and contextual factors, such as age, stimulus modality, or task could favor the conscious negativity bias expression [1]. For example, stress can increase the bias level [9]. It produces a relocation of resources in the brain, favoring subcortical regions activity over the prefrontal cortex [41]. In this situation, top-down processes may not function properly, and negativity bias may be more easily manifested at the conscious response. Therefore, it will be necessary to explore a wide range of factors to understand when we are particularly vulnerable to this bias.

The fact that the bias only appeared at the implicit level when judging ambiguous faces does not imply that it could not be influencing other cognitive domains. Thus, regarding risky decision-making, results confirmed the hypothesis that negativity bias would be conducive to more biased decisions. Specifically, the greater the unconscious attraction towards negative ratings when classifying ambiguous faces as positive, the higher the loss aversion. These results would be in line with Sheng et al. [24], who highlighted that loss aversion could be explained, at least partially, by the negativity bias. In addition, authors found that this bias was unconsciously manifested through increased visual attention to losses. This may also fit with our results, which showed that only the implicit negativity bias was significantly associated with loss aversion. Nevertheless, to clarify these issues, it would be necessary to address, through instruments such as an eye-tracker [42], if the negativity bias measure used in this study could be also related to the heightened focus on losses reported by Sheng et al. [25], as well as whether it is behaviorally meaningful, influencing other complex decisions as reflected in recent studies, where even the predisposition to adopt innovative technologies would depend on the level of negativity bias [13].

On the other side, regarding decision-making under ambiguity, our last hypothesis stated that negativity bias would increase the IGT performance since this bias would favor reinforcement-learning. The overall score was similar for the different levels of the negativity bias; however, in line with our hypothesis, this bias was associated with faster learning and greater performance through the task. Specifically, those who most automatically rated ambiguous faces as negative performed better in the second and third blocks of the IGT and showed a trend towards better performance in the fourth block. Since our sample was composed of healthy, young participants and they should not face difficulties in learning the appropriate strategy in IGT [33], the margin for improvement attributed to the negativity bias may not be large enough to be observed in the overall score. However, studying the learning curve through the different blocks allowed for further exploration.

According to Bechara et al. [43], during the pre-punishment period (first block), participants do not know how the task works and must explore. Therefore, negativity bias could not explain their performance as the choices would be random. However, during the second and third blocks, called hunch periods, participants begin to develop anticipatory emotional signals based on their experiences [27,29] and their sensitivity to feedback [20]. Since the negativity bias would help to focus attention on negative information [3,30], it could help to generate such anticipatory markers and facilitate the avoidance of disadvantageous decks, improving the performance, as our results showed. Finally, Brand et al. [44] argued that the last blocks are less ambiguous, and participants rely on the attributions developed during the task. Yet, these attributions may be affected by multiple factors such as personality, working-memory, and impulsiveness, among others [45,46]. Thus, negativity bias could become particularly important only in ambiguous phases where it is still difficult to decide based on conscious information. Nevertheless, more research is needed to verify whether the bias really becomes secondary when participants form their hypotheses. In this line, Bechara et al. [43] studied participants’ attributions throughout the task by asking them at the end of each block about their beliefs. It would be useful to replicate this approach in future studies also addressing negativity bias.

This study is not exempt from limitations, mainly related to potential variability of the negativity bias. Firstly, it was found that men would have a lower negativity bias [47]. Although the role of sex was considered by adjusting results by sex, the disproportionate sample (mostly women) makes it difficult to draw conclusions. On the other hand, all participants were young. Authors such as Carstensen & DeLiema [48] suggested that the negativity bias present in youth would decrease with age. Moreover, the measure of the bias was based on only one type of stimuli (emotional faces) and may differ if addressed with others [1]. Thus, it would be appropriate to replicate our study with a proportionate sample of men and women, covering different ages, and using different measures of negativity bias, to check if results can be generalized.

## 5. Conclusions

Our work highlights that the same bias could lead to different results depending on the context. In risky contexts, under the classical-rationality framework [19], it could be concluded that negativity bias is leading to less rational decisions, which are often interpreted as negative. In fact, these approaches have resulted in libertarian paternalism policies [49] that consider we need a “nudge” [50] to avoid biases that affect us when deciding. However, negativity bias could favor good decisions in ambiguous contexts such as the IGT. Here, this bias could act as an enhancer of reinforcement-learning by providing greater sensitivity to punishment, which would help to avoid future negative consequences. In fact, from evolutionary perspectives, this bias represents an adaptive advantage that errs on the side of caution, maximizing survival [51,52]. But again, this would depend on the context. The IGT is designed to “reward” caution but in an ambiguous task that rewards risk-taking, negativity bias would be negative once again. Our data therefore seem to support the ecological rationality approach [17,18] and the need to contextualize rather than draw general conclusions about whether a phenomenon is inherently good or bad. As Simon [53] stated: “Human rational behavior is shaped by a scissors whose blades are the structure of task environments and the computational capabilities of the actor”. It is important that, in the future, the scientific community properly explores the role of biases, rather than simply criticizing them, as in some contexts they may even be a useful tool for making good decisions faster and at lower cost.

## Figures and Tables

**Figure 1 ijerph-19-17037-f001:**
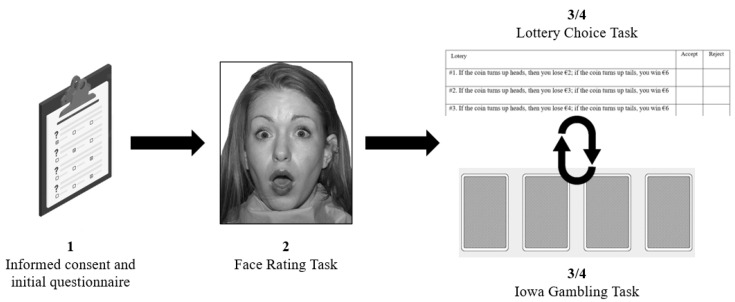
Experimental session procedure. The order in which the tasks were performed during the experimental session is shown. The Iowa Gambling Task and Lottery Choice Task were counterbalanced across participants.

**Figure 2 ijerph-19-17037-f002:**
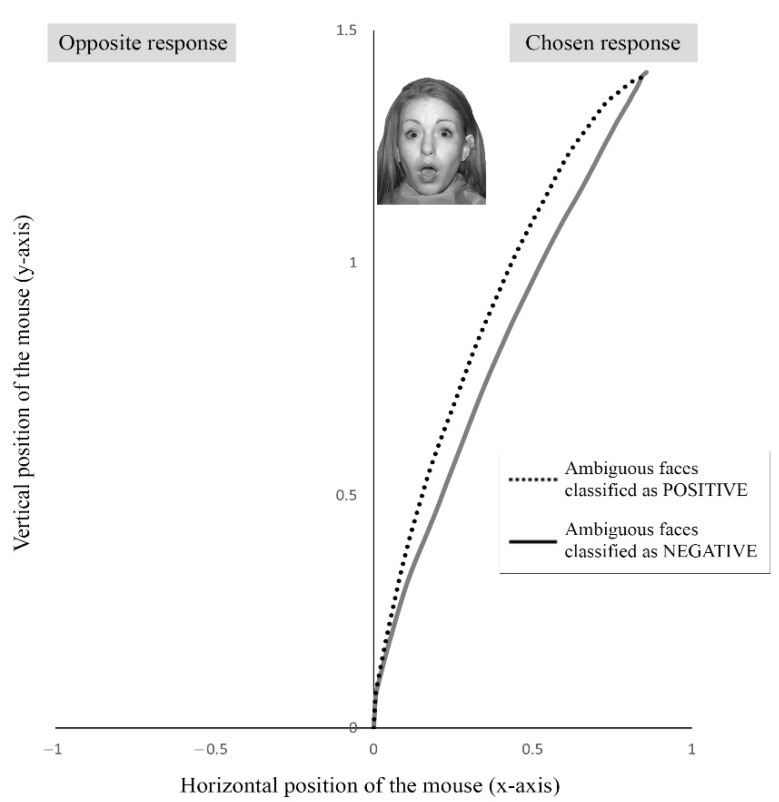
Mouse trajectory when classifying ambiguous faces as negative or positive. There was a greater maximum deviation of the mouse when classifying ambiguous faces as “positive” than “negative”. When participants classified ambiguous faces as “positive”, they showed response trajectories that indicated a greater attraction towards the competitive option (negative), as opposed to when they classified an ambiguous face as “negative”, which they did more automatically.

**Figure 3 ijerph-19-17037-f003:**
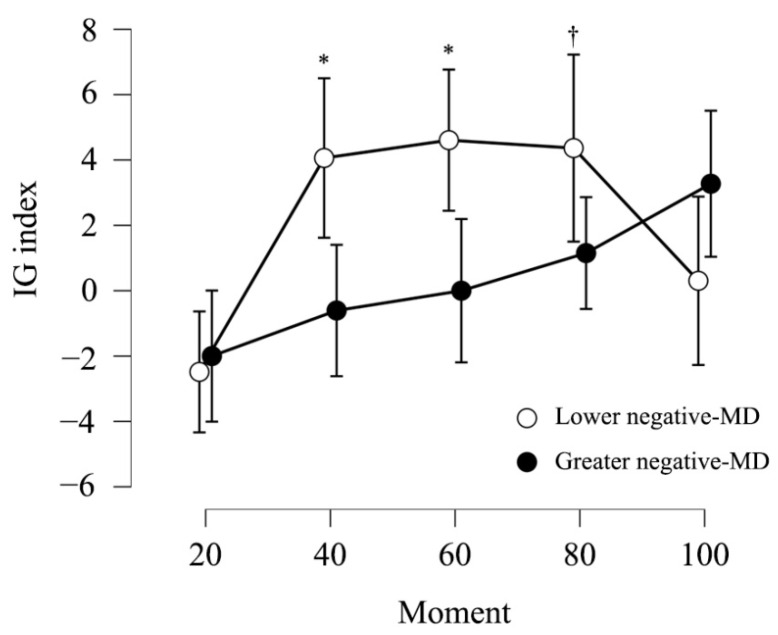
Performance in the IGT blocks depending on the group (higher or lower negative-MD). Participants who classified surprise or ambiguous faces as negative more directly, i.e., showed more negative bias at the implicit level, performed significantly better (*) in blocks 2 and 3 of the IGT. In addition, they showed a trend (†) towards better performance in block 4. Means ± 95% confidence interval.

**Table 1 ijerph-19-17037-t001:** Inter-subject effect tests for the different IGT blocks.

		Lower Negative-MD (*N* = 34)	Greater Negative-MD (*N* = 35)	*F*	Gl Hypothesis	Gl Error	*p*-Value	η2p
IGT	Block 1	*M* = −2.48 ± 5.22	*M* = −2.00 ± 5.65	0.13	1	64	0.719	0.002
	Block 2	*M* = 4.06 ± 6.88	*M* = −0.61 ± 5.66	9.04	1	64	0.004 **	0.124
	Block 3	*M* = 4.60 ± 6.09	*M* = 0.00 ± 6.18	9.28	1	64	0.003 **	0.127
	Block 4	*M* = 4.36 ± 8.06	*M* = 1.15 ± 4.82	3.85	1	64	0.054 ^†^	0.057
	Block 5	*M* = 0.30 ± 7.265	*M* = 3.27 ± 6.30	3.14	1	64	0.081	0.47

IGT, Iowa Gambling Task; *M*, mean; ± standard deviation; **** significant contrast at the 0.01 level. ^†^ significant trend.

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
