# Peer review of "Implicit Negativity Bias Leads to Greater Loss Aversion and Learning during Decision-Making"

_ijerph, 2022, doi:10.3390/ijerph192417037_

Round 1

Reviewer 1 Report

I think you have put forward a research that can contribute to the literature. I have some suggestions that I think you can contribute to your research;

1. A flowchart can be added showing all the stages of conducting the research. This will provide semantic convenience for readers.

2. It may be useful for your research to review and cite the publication I have mentioned below.

https://www.acrwebsite.org/volumes/6335/

3. You used Mouse Tracker software. If you have obtained permission from the participants in the consent form, you can combine the screenshot of the experiment and the images of the participants during the experiment and present them as a single image.

4. In future versions of your research, you can use brain signals, facial line emotion extraction algorithms and sensors such as GSR, EKG, Pulsemeter to collect data. Generating different versions of your research with multimodal signal processing will contribute to the literature.

5. In future research, you can create a database of your own country and culture instead of using a ready-made image-emotion database.

Author Response

I think you have put forward research that can contribute to the literature. I have some suggestions that I think you can contribute to your research.

  1. A flowchart can be added showing all the stages of conducting the research. This will provide semantic convenience for readers.

Following your suggestion, we have developed a flowchart to accompany the procedure section to facilitate the understanding of the protocol and improve the readability of the study.

  1. It may be useful for your research to review and cite the publication I have mentioned below. https://www.acrwebsite.org/volumes/6335/

The review you suggest has been of particular relevance to us and has been incorporated into the introduction of this study. On the one hand, by inviting readers to learn more about the theoretical proposals on the origin of the negativity bias and, on the other hand, by expanding our view that Kahneman was the only author to connect this bias with loss aversion. Thus, this connection is also theorised in Kanouse's review and it is important to highlight this.

  1. You used Mouse Tracker software. If you have obtained permission from the participants in the consent form, you can combine the screenshot of the experiment and the images of the participants during the experiment and present them as a single image.

What you suggest is a very good idea which would certainly add to the attractiveness of our work and help to illustrate it. Unfortunately, this was not considered during the research and no photos were taken of the participants during the session, just as this was not included in the informed consent. However, as we intend to continue using this software in the future, we will consider including it in future studies.

  1. In future versions of your research, you can use brain signals, facial line emotion extraction algorithms and sensors such as GSR, EKG, Pulsometer to collect data. Generating different versions of your research with multimodal signal processing will contribute to the literature.

This is a very good suggestion that we will certainly take into account for future studies. Obtaining multimodal data will allow us to strengthen our findings and even broaden our knowledge by exploring possible neural and physiological bases of the negativity bias.

  1. In future research, you can create a database of your own country and culture instead of using a ready-made image-emotion database.

Finally, we will also take this suggestion into account. In the first instance we did not do it as you mention because we wanted to test specific hypotheses and we were concerned that, if they were not fulfilled, we would not know whether it was perhaps because we were using images that had not been previously validated. We therefore chose to use a standardised and pre-tested battery of images. However, for future versions it would be interesting to include photos with other elements typical of the culture of this country, and even to test the hypothesis of whether the results are replicated depending on whether the stimuli are typical of the culture or whether they come from other cultures outside our own.

Reviewer 2 Report

The manuscript is well written. Introduction, materials and methods, results and discussion are clear and understandable. English language requires minor corrections to be a formal one.

Author Response

We really appreciate the time you have taken to review our work and we are pleased to receive your positive feedback. Regarding the English, the current manuscript has been completely revised to improve on the shortcomings.

Reviewer 3 Report

1.     In the introduction and discussion part, the authors need to include more recent studies such as for the years 2022 and 2021 with a proper logical sequence.

2.     I am not clear about the objective and significance of this study compared to existing studies, please clarify it.

3.     From section 2 (Materials and Methods), I am not clear about the statistical sampling method the authors have used for selecting participants, please explain with clarity. I am also confused about how authors are confident about their sample size. Have you used any validation tests based on your selected sample size?

4. The author has shown two figures in this article, but I cannot see any title for these figures, please include the title.

5. How your research may be helpful for the future work of the scientific community, clarify your reasons.

Author Response

  1. In the introduction and discussion part, the authors need to include more recent studies such as for the years 2022 and 2021 with a proper logical sequence.

Following your suggestion, we have reviewed and included during the introduction and discussion some more recent works that we had not previously considered (such as Norris, 2021) and that, in fact, have been relevant and we believe that they strengthen the theoretical body of our study. However, if you feel that we have missed any relevant references, please let us know and we will update it again.

  1. I am not clear about the objective and significance of this study compared to existing studies, please clarify it.

We have added further explanation of the aims and relevance of this study in the introduction and, following your subsequent commentary (number 5) on what it brings to the scientific community, this has also been developed in the conclusion. We hope that you find the current version of the manuscript adequate and clearer, however, we would be happy to take up further suggestions and make further clarifications if you feel this is appropriate.

  1. From section 2 (Materials and Methods), I am not clear about the statistical sampling method the authors have used for selecting participants, please explain with clarity. I am also confused about how authors are confident about their sample size. Have you used any validation tests based on your selected sample size?

In response to your comment, the current version of the manuscript details the sample collection method (non-probabilistic method by convenience), as well as the power calculation we performed a priori, to ensure an adequate sample size.

  1. The author has shown two figures in this article, but I cannot see any title for these figures, please include the title.

The titles were actually written, but perhaps they were not detailed and could not be seen properly. We have underlined the titles in yellow in the current revised version.

  1. How your research may be helpful for the future work of the scientific community, clarify your reasons.

As mentioned in the response to comment 2, we have added some comments in the conclusions that may clarify the relevance of this study for the scientific community. Again, we hope you like it, but if you think we should modify or add anything else, please let us know.

Round 2

Reviewer 3 Report

Thank you all authors for updating the article according to the suggestions given in the first review.